# piRNA-guided co-transcriptional silencing coopts nuclear export factors

Martin H Fabry[†], Filippo Ciabrelli[†], Marzia Munafo[†], Evelyn L Eastwood, Emma Kneuss, Ilaria Falciatori, Federica A Falconio, Gregory J Hannon*, Benjamin Czech*

Cancer Research UK Cambridge Institute, University of Cambridge, Cambridge, United Kingdom

**Abstract** The PIWI-interacting RNA (piRNA) pathway is a small RNA-based immune system that controls the expression of transposons and maintains genome integrity in animal gonads. In *Drosophila*, piRNA-guided silencing is achieved, in part, via co-transcriptional repression of transposons by Piwi. This depends on Panoramix (Panx); however, precisely how an RNA binding event silences transcription remains to be determined. Here we show that Nuclear Export Factor 2 (Nxf2) and its co-factor, Nxt1, form a complex with Panx and are required for co-transcriptional silencing of transposons in somatic and germline cells of the ovary. Tethering of Nxf2 or Nxt1 to RNA results in silencing of target loci and the concomitant accumulation of repressive chromatin marks. Nxf2 and Panx proteins are mutually required for proper localization and stability. We mapped the protein domains crucial for the Nxf2/Panx complex formation and show that the amino-terminal portion of Panx is sufficient to induce transcriptional silencing.
DOI: https://doi.org/10.7554/eLife.47999.001

*For correspondence:
greg.hannon@cruk.cam.ac.uk
(GJH);
benjamin.czech@cruk.cam.ac.uk
(BC)

[†]These authors contributed equally to this work

Competing interests: The authors declare that no competing interests exist.

## Introduction

The piRNA pathway is a small RNA-based immune system that represses transposable elements in animal gonadal tissues (*Czech et al., 2018*; *Ozata et al., 2019*). At the core of this pathway are PIWI-clade Argonaute proteins that are guided by 23-30nt piRNA partners to silence transposon targets via two main mechanisms. In *Drosophila*, Aubergine and Argonaute 3 enforce post-transcriptional gene silencing (PTGS) via direct cleavage of transposon mRNAs in the cytoplasm (*Brennecke et al., 2007*; *Gunawardane et al., 2007*). Piwi, in contrast, operates in the nucleus where it instructs the co-transcriptional gene silencing (TGS) of transposon insertions (*Brennecke et al., 2007*; *Klenov et al., 2011*; *Sienski et al., 2012*). Mutations that compromise TGS result in severe loss of transposon control, despite normal piRNA levels (*Dönertas et al., 2013*; *Le Thomas et al., 2013*; *Muerdter et al., 2013*; *Ohtani et al., 2013*; *Rozhkov et al., 2013*; *Sienski et al., 2015*; *Sienski et al., 2012*; *Yu et al., 2015*).

Piwi, in complex with piRNAs, detects nascent transposon RNAs arising from active insertions and directs the silencing of these loci. Target silencing is achieved via recruitment of histone modifying enzymes that deposit repressive chromatin marks, mainly trimethylation of Lysine 9 on Histone 3 (H3K9me3) (*Iwasaki et al., 2016*; *Klenov et al., 2014*; *Le Thomas et al., 2013*; *Rozhkov et al., 2013*; *Sienski et al., 2012*; *Wang and Elgin, 2011*). Panoramix (Panx) is a key TGS effector, acting downstream of Piwi at the interface between the piRNA pathway and the general chromatin silencing machinery (*Sienski et al., 2015*; *Yu et al., 2015*). Strikingly, RNA-mediated recruitment of Panx, but not Piwi, to a locus is sufficient to trigger its epigenetic silencing, thus placing Panx at a critical node of the TGS mechanism. Downstream of Panx, the concerted action of dLsd1/Su(var)3–3 and Eggless/dSETDB1 erases H3K4me2 and concomitantly deposits H3K9me3, followed by chromatin compaction via Heterochromatin Protein 1a (HP1a/Su(var)205) (*Czech et al., 2013*; *Iwasaki et al.,*

**eLife digest** For an organism to form and grow correctly, it must rely on the genetic information it has received from its parents. DNA, however, is full of elements called transposons that can disrupt this information by moving around the genome and inserting themselves into genes. Changes caused by these invading elements can lead to devastating effects, such as cancer and cell death. To shield their DNA from harm, organisms have evolved regulatory machineries to recognize and correct alterations that may be damaging.

One way cells can protect their DNA is by silencing disruptive transposons using small molecules known as piRNAs. These protective molecules detect transposons as soon as they are active and recruit other proteins to switch them off. However, questions still remain about how specific proteins recruited by piRNAs are involved in this process. In flies, a protein called Panoramix (Panx) is known to trigger transposon silencing, but how it does this is still unclear.

Now, Fabry, Ciabrelli, Munafò et al. set out to investigate how Panx silences active transposons in the ovaries of flies and whether other proteins are involved. Like so many other proteins, Panx was found not to work alone but to form a complex with two other proteins, called Nxf2 and Nxt1. The experiments showed that all three components of the complex, named PICTS, are critical for transposon control in flies, but Panx is the engine that drives the machine.

Panx and Nxf2 were found to stabilize each other, and together with co-factor Nxt1 place a mark on the genome at the point where the transposon emerges, effectively switching it off. Notably, although Nxf2 and Nxt1 are part of a family of proteins that export molecules from the nucleus, both these factors appear to have been repurposed to silence transposable elements within the genome.

This work expands our understanding of how cells employ regulatory machineries, like the PICTS complex, to guard against disruptive genetic changes. These mechanisms are often conserved throughout evolution, and the findings presented here may help identify ways to counteract harmful changes caused by transposons in other organisms, including humans. However, more work would be required to deepen our knowledge of how these processes work.
DOI: https://doi.org/10.7554/eLife.47999.002

*2016*; *Rangan et al., 2011*; *Sienski et al., 2015*; *Wang and Elgin, 2011*; *Yu et al., 2015*). Precisely how Panx recruits these histone modifying enzymes and what other factors participate in this process remains an outstanding question.

Here we show that Panx coopts elements of the nuclear RNA export machinery to trigger transcriptional silencing. Panx is part of a complex that also contains Nuclear Export Factor 2 (Nxf2) and Nxt1/p15. Panx and Nxf2 are interdependent for their protein stability. *nxf2* mutants show strong de-repression of Piwi-regulated transposons and severe loss of H3K9me3 at affected loci, similarly to *panx* mutants. We find that the amino-terminus of Panx delivers the critical silencing signal, as it is necessary and sufficient to trigger the deposition of repressive chromatin marks if tethered to a reporter construct, while its carboxyl-terminal region is involved in the interaction with Nxf2. Nxf2 is closely related to the mRNA export factor Nxf1, which also interacts and functions with Nxt1 (*Fribourg et al., 2001*; *Herold et al., 2001*; *Herold et al., 2000*). Thus, our findings reveal that the evolution of transposon defense mechanisms involved exaptation of the nuclear RNA export machinery.

## Results

### Nxf2 is a TGS factor that interacts with Panx

To identify proteins associated with Panx, we immunoprecipitated a GFP-Panx fusion protein expressed from its endogenous promoter (*Handler et al., 2013*; *Figure 1—figure supplement 1A*) from ovary lysates and identified co-purifying proteins by quantitative mass spectrometry. Three proteins showed the strongest enrichment and significance: Panx, Nxf2 and Nxt1 (*Figure 1A*, *Figure 1— source data 1*). Nxf2 is a homolog of the general messenger RNA (mRNA) export factor Nxf1 but was reported previously as being dispensable for canonical mRNA transport in S2 cells

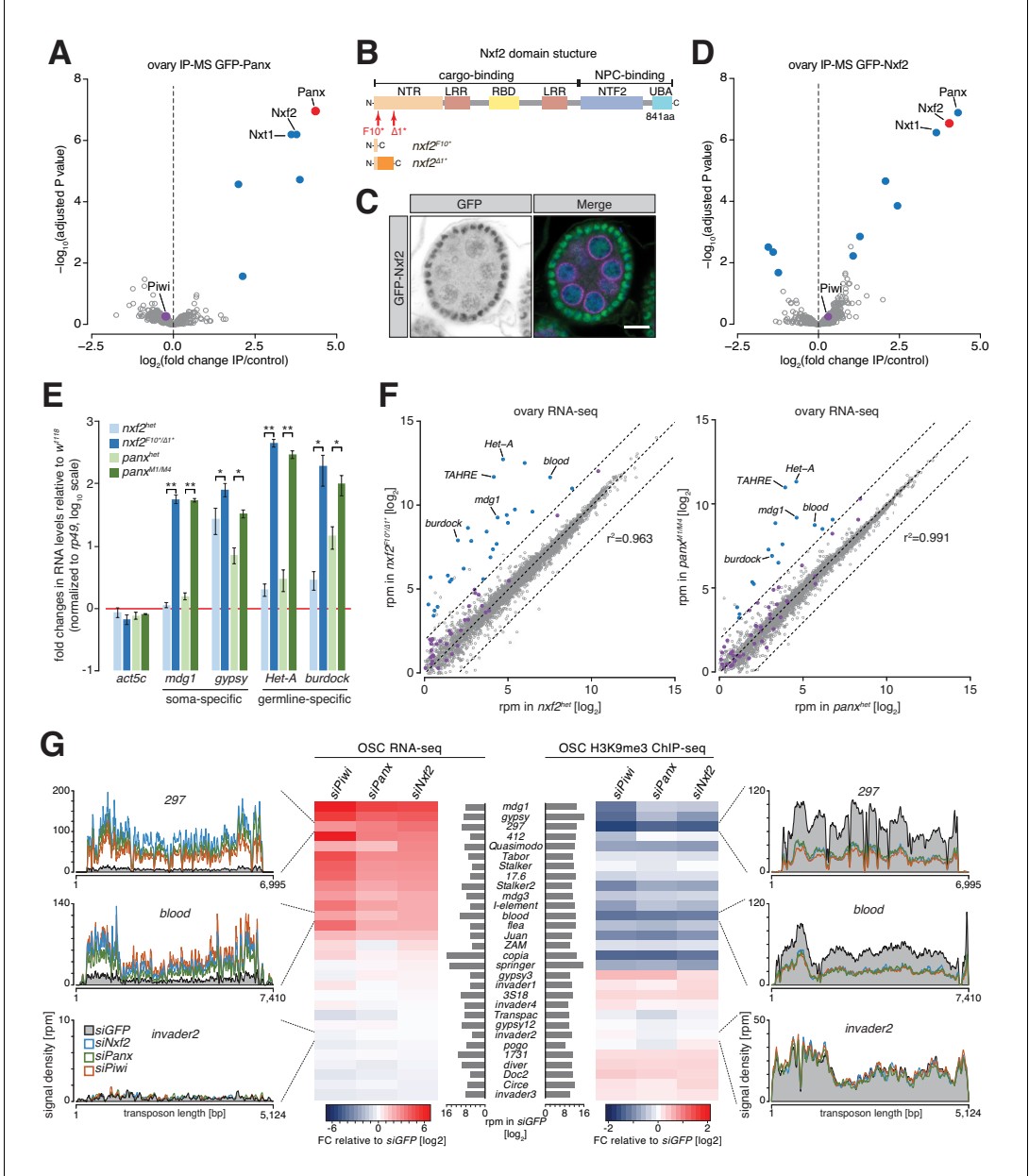

**Figure 1.** Nxf2 is a piRNA pathway factor that functions in transcriptional gene silencing. (**A**) Volcano plot showing enrichment values and corresponding significance levels for proteins co-purified with GFP-Panx from ovary lysates (n = 4 for GFP-Panx and n = 3 for control ovaries). Proteins with fold change >2 and adjusted P value < 0.05 are highlighted in blue. The bait protein is labeled in red and Piwi is shown in purple. (**B**) Cartoon displaying the Nxf2 domain structure and generated mutants. NTR, amino-terminal region; LRR, leucine rich repeats; RBD, RNA-binding domain; NTF2, NTF2-like domain; UBA, Ubiquitin associated domain. (**C**) Expression and localization of GFP-Nxf2 in an egg chamber is shown by immunofluorescence (see also *Figure 1—figure supplement 1C*). Green, GFP-Nxf2; magenta, Aubergine; blue, DNA. Scale bar, 10 µm. (**D**) as (**A**) but co-purification from GFP-Nxf2 ovary lysates (GFP-Nxf2, n = 4; control, n = 3). (**E**) Bar graphs showing fold changes in steady-state RNA levels of soma- (*mdg1*, *gypsy*) and germline- (*HeT-A*, *burdock*) specific transposons in total ovarian RNA from the indicated genotypes (relative to wild-type and normalized to *rp49*). * denotes P value < 0.05; ** denotes p<0.001 (unpaired t-test). Error bars indicate standard deviation (n = 3). (**F**) Scatter plots showing expression levels (reads per million sequenced reads) of genes (in grey) and transposons (in purple) from total RNA from ovaries of the indicated genotypes (left, *nxf2*; right, *panx*; n = 3; $r^2$ values represent expression of genes only). Transposons whose abundance change more than four-fold compared to heterozygotes are highlighted in blue. (**G**) Heat maps showing RNA-seq (left) and H3K9me3 ChIP-seq (right) of the 30 most expressed transposons in OSCs (compared with *siGFP*) upon the indicated knockdowns. Density profiles of normalized reads from RNA-seq (left) and H3K9me3 ChIP-seq (right) experiments mapping to the indicated transposons.

DOI: https://doi.org/10.7554/eLife.47999.003

*Figure 1 continued on next page*

*Figure 1 continued*

The following source data and figure supplements are available for figure 1:

**Source data 1.** List of proteins recovered in GFP-Panx IP-MS.
DOI: https://doi.org/10.7554/eLife.47999.006
**Source data 2.** List of proteins recovered in GFP-Nxf2 IP-MS.
DOI: https://doi.org/10.7554/eLife.47999.007
**Figure supplement 1.** Nxf2 is a piRNA pathway factor acting in TGS.
DOI: https://doi.org/10.7554/eLife.47999.004
**Figure supplement 2.** Loss of Nxf2 impairs TGS at levels comparable to Panx.
DOI: https://doi.org/10.7554/eLife.47999.005

(*Herold et al., 2001*; *Herold et al., 2003*). Nxf2 contains all domains present in the family defined by Nxf1, namely an amino-terminal region (NTR), an RNA-binding domain (RBD), leucine-rich repeats (LRR), the NTF2-like domain (NTF2) and a Ubiquitin-associated (UBA) domain (*Figure 1B* and *Figure 1—figure supplement 1B*) (*Herold et al., 2001*). While the NTR, LRRs and RBD are typically involved in cargo binding, the NTF2 and UBA domains mediate binding to the Nuclear Pore Complex (NPC) and are required for Nxf1-mediated RNA export (*Braun et al., 2001*; *Fribourg et al., 2001*). Nxt1, also known as p15, is a co-factor of Nxf1 responsible for interaction with the NPC, specifically through the NTF2-fold (*Fribourg et al., 2001*; *Lévesque et al., 2001*). Interestingly, Nxt1 was also reported to interact with Nxf2 (*Herold et al., 2001*; *Herold et al., 2000*). Both Nxf2 and Nxt1 were previously identified in screens for piRNA-guided silencing in somatic and germline cells, and their depletion resulted in female sterility (*Czech et al., 2013*; *Handler et al., 2013*; *Muerdter et al., 2013*). Contrary to expectations based upon previous findings (*Sienski et al., 2015*; *Yu et al., 2015*), we saw no enrichment for Piwi by mass spectrometry (*Figure 1A*), results that are consistent with another recent study (*Batki et al., 2019*). However, co-immunoprecipitation experiments detected weak but reproducible interactions between Piwi and Panx, Nxf2, and Nxt1, but not with a negative control (*Figure 1—figure supplement 1D*), suggesting that low amounts of transposon substrates in unperturbed cells and/or transient associations might push Piwi below the limit of detection by less sensitive approaches.

Using CRISPR/Cas9, we generated flies that express a GFP-Nxf2 fusion protein from the endogenous *nxf2* locus. GFP-Nxf2 is expressed in follicle and germline cells of the ovary and localizes predominantly to nuclei (*Figure 1C* and *Figure 1—figure supplement 1C*). Mass spectrometry of GFP-Nxf2-associated proteins identified Panx, Nxf2, and Nxt1, as binding partners (*Figure 1D*, *Figure 1—source data 2*), implying the existence of a complex containing these three factors, which we named the <u>P</u>anx-<u>i</u>nduced <u>c</u>o-<u>t</u>ranscriptional <u>s</u>ilencing (PICTS) complex.

We therefore generated two nxf2 mutant alleles, *nxf2^{F10*}* and *nxf2^{Δ1*}*, which harbor premature stop codons that disrupt the *nxf2* open reading frame from amino acid 10 onwards (*Figure 1B* and *Figure 1—figure supplement 1E*). Trans-heterozygous mutants were female sterile, with fewer eggs laid and none hatching (*Figure 1—figure supplement 1F*). *nxf2* mutants were severely compromised in the repression of soma- and germline-specific transposons, in a highly similar manner to *panx* mutants (*Figure 1E*), with no change in piRNA levels or Piwi localization, despite compromised silencing (*Figure 1—figure supplement 1G–H*).

To assess the specificity of the impact of *nxf2* mutations on the transcriptome, we performed RNA-seq from total RNA of heterozygote and mutant ovaries, using *panx* mutants for comparison. As reported previously, the expression of protein-coding genes was not generally affected in *nxf2* mutants, with only 16 out of 7252 ($r^2$ = 0.963) being changed more than 4-fold (*Figure 1F*) (*Herold et al., 2001*; *Herold et al., 2003*). In contrast, 28 out of 60 transposon families (that were above the expression threshold of 1 rpm) were de-repressed by more than 4-fold in *nxf2* mutants. Similar results were obtained for *panx* mutants: 16 out of 60 transposons were de-repressed and only 6 out of 7252 genes mis-regulated ($r^2$ = 0.991). ChIP-seq for the H3K9me3 mark from *nxf2* mutant ovaries showed reduced methylation levels at the same transposon families that were de-repressed according to RNA-seq, such as *Het-A*, while randomly chosen genomic intervals were not changed (*Figure 1—figure supplement 2A–B*).

Ovarian somatic cells (OSCs), cultured in vitro, express a functional piRNA-guided co-transcriptional gene silencing machinery and provide a convenient context for mechanistic studies

(*Saito et al., 2009*). RNA-seq from OSCs depleted of Piwi, Panx or Nxf2 showed marked de-repression of soma-specific (e.g. *mdg1, gypsy, 297*) and intermediate transposon families (e.g. *blood*) when compared to control cells treated with *GFP* siRNAs (*Figure 1G* left; *Figure 1—figure supplement 2C*). The de-repression of transposon families strongly correlates with the reduction in H3K9me3 levels mapped over their consensus sequences in ChIP-seq samples generated from the same knockdowns (*Figure 1G* right). In contrast, no major changes in H3K9me3 were detected over transposons that show no de-repression in these cells upon *piwi*, *panx*, or *nxf2* knockdown.

We next focused on a set of 233 individual, Piwi-regulated genomic transposon insertions in OSCs (see Materials and methods for details). This enabled analysis of chromatin states on individual loci, including flanking regions, rather than averaging contributions over a consensus sequence (*Figure 1—figure supplement 2D–H*). Piwi depletion resulted in the accumulation of H3K4me2 at these loci and spreading of the mark, indicative of active transcription, beyond the transposon into downstream regions (*Figure 1—figure supplement 2D,F*), similar to earlier reports (*Dönertas et al., 2013*; *Klenov et al., 2014*; *Sienski et al., 2012*). Knockdown of *panx* or *nxf2* showed similar, though less pronounced, effects. H3K9me3 marks were strongly reduced upon Piwi depletion, with *panx* and *nxf2* knockdowns showing a similar but milder impact (*Figure 1—figure supplement 2E,G*). H3K4me2 spreading typically correlates with increased RNA output and a decrease in H3K9me3 levels, as evident for a euchromatic *gypsy* insertion located within an intron of the 5' UTR of the gene *ex* on chromosome 2L (*Figure 1—figure supplement 2H*).

## Panx and Nxf2 proteins are interdependent for their stability

Proteins that form complexes are often interdependent for either localization or stability, and there are abundant examples of such interactions in the piRNA pathway (*Dönertas et al., 2013*; *Ohtani et al., 2013*). To test for such dependencies among TGS factors, we depleted Piwi, Panx or Nxf2 in germ cells of flies expressing GFP-Nxf2 (*Figure 2A*) or GFP-Panx (*Figure 2B*). Germline knockdown of *piwi* had no effect on the localization of either Nxf2 or Panx. Depletion of Panx, however, led to a pronounced loss of nuclear GFP-Nxf2 in nurse cell nuclei (*Figure 2A*). The reciprocal was also true, with GFP-Panx nuclear signal being reduced upon *nxf2* knockdown in nurse cells (*Figure 2B*). Similarly, the individual depletion of Panx or Nxf2 in follicle cells resulted in a reduction of both proteins (*Figure 2—figure supplement 1A–B*). To assess whether the observed reduction reflects protein stability rather than mislocalization, we performed western blots on ovarian lysates. Panx protein level was strongly reduced in *nxf2* mutant ovaries (*Figure 2C*) and GFP-Nxf2 signal was completely lost in homozygous *panx* mutants (*Figure 2D*). Of note, mRNA levels of Nxf2 and Panx were not affected when the other factor was mutated (*Figure 1F*), implying regulation at the protein level. Considered together, the localization and stability of Nxf2 and Panx are interdependent in vivo.

## PICTS complex formation is required for TGS

To map the domains of Nxf2 and Panx that mediate their interaction, we expressed various combinations of full-length, truncated, or mutant proteins in S2 cells or in OSCs where native protein expression had been reduced by RNAi (*Figure 3A*). Interactions were tested by co-immunoprecipitation and western blot analyses, and the subcellular localization analyzed by immunofluorescence staining. In OSCs, the ability of each construct to rescue transposon de-repression was monitored by qPCR.

Full-length Nxf2 and Panx robustly co-immunoprecipitated (*Figure 3B*) and co-localized in S2 cell nuclei (*Figure 3C*). Removing the carboxy-terminal half of Panx (Panx-ΔC) yielded a protein that remained nuclear, while co-expressed Nxf2 remained largely cytoplasmic, and these proteins no longer formed a complex (*Figure 3B–D*). When expressed alone in S2 cells, Nxf2 remained predominantly cytoplasmic (ZsGreen-HA in *Figure 3C*), suggesting that interaction with Panx is necessary for nuclear localization of Nxf2. Panx-ΔN, in contrast, retained the ability to interact with Nxf2 but failed to localize to the nucleus (*Figure 3B–D*). Strikingly, enforced localization of Panx-ΔN to the nucleus also restored nuclear localization of Nxf2 (*Figure 3D* and *Figure 3—figure supplement 1C–D*). Deleting only either the coiled coil domain #2 or C-terminal region, which together make up most of the carboxy-terminal half of Panx, reduced co-purification with Nxf2 (*Figure 3D* and *Figure 3—figure supplement 1C*), with neither construct being able to rescue *mdg1* repression upon *panx*

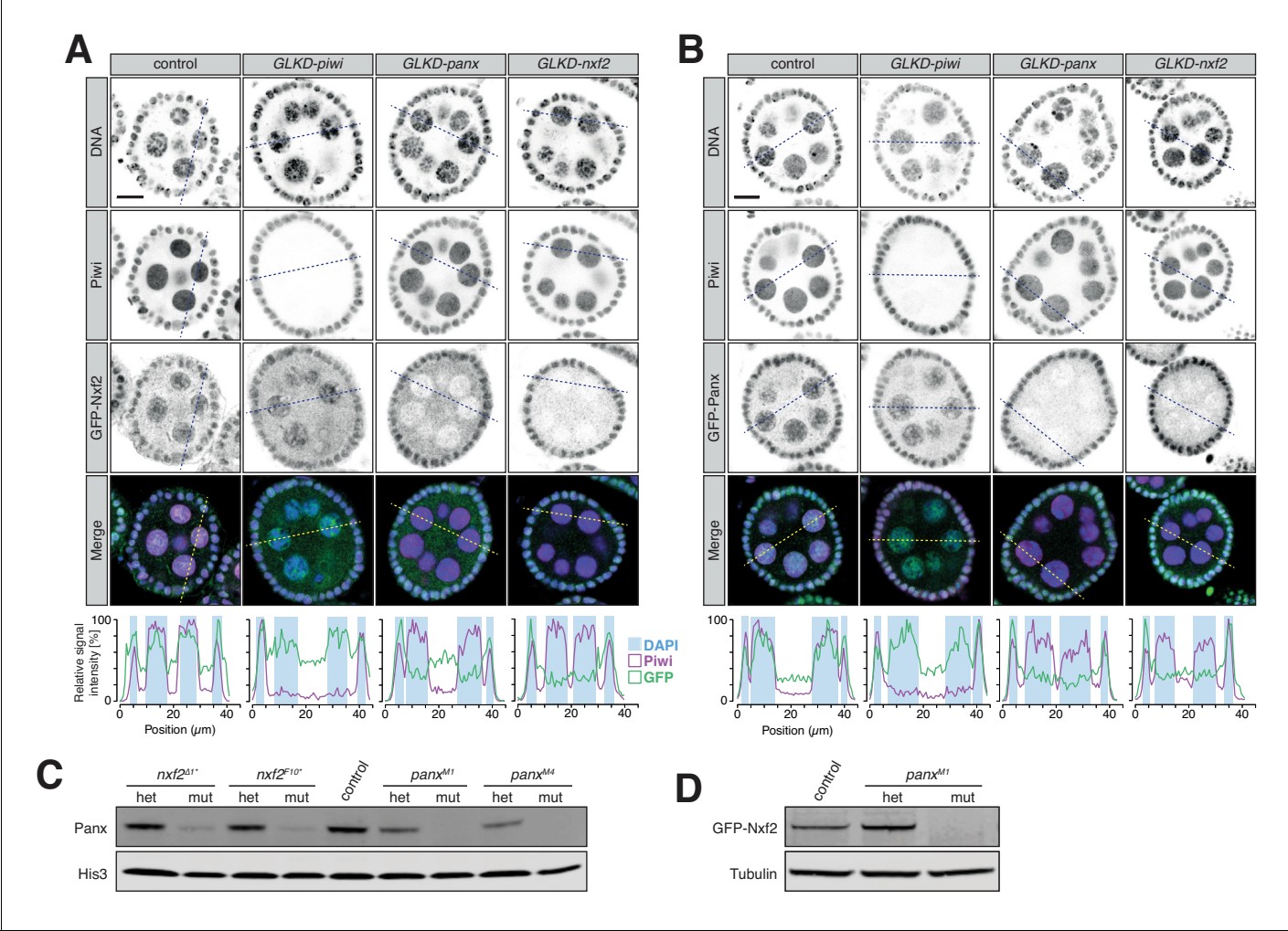

**Figure 2.** Protein stability and localization of Nxf2 and Panx is reciprocally co-dependent. (A) Expression and localization of Piwi and GFP-Nxf2 in egg chambers upon germline-specific knockdown (GLKD) of the indicated factors are shown by immunofluorescence (GFP, green; Piwi, magenta; DNA, blue). Scale bar, 10 μm. Plot profiles showing the relative signal intensity of Piwi (magenta), GFP-Nxf2 fusion protein (green) and DAPI (blue background box) were computed from the indicated sections. (B) as (A) but showing GFP-Panx. (C) Western blot showing Panx protein levels in ovaries from *panx* and *nxf2* heterozygotes and mutants compared to $w^{1118}$ control flies. His3 served as loading control. (D) as (C) but showing levels of GFP-Nxf2 protein.
DOI: https://doi.org/10.7554/eLife.47999.008

The following figure supplement is available for figure 2:

**Figure supplement 1.** Protein stability and localization of Nxf2 and Panx is reciprocally co-dependent.
DOI: https://doi.org/10.7554/eLife.47999.009

knockdown (*Figure 3E*). Overall, these results suggest that the N-terminal part of Panx carries its nuclear localization signal that aids proper localization of Nxf2 via interaction with the Panx C-terminal region.

To probe the regions of Nxf2 that are essential for its function, we expressed, in the presence of full-length Panx, Nxf2 proteins that lacked either the regions required for RNA cargo binding or the region essential for its association with the NPC (*Braun et al., 2001*; *Fribourg et al., 2001*; *Herold et al., 2001*) (*Figure 3A* and *Figure 3—figure supplement 2A* left). Nxf2-ΔNPC failed to co-purify with Panx, and this was accompanied by an increased cytoplasmic protein localization (*Figure 3B–D*). In contrast, deleting the cargo-binding region of Nxf2 had less impact on its co-purification with Panx or the nuclear localization of either protein (*Figure 3B–D*). This mutant was able to interact with Panx-ΔN but not Panx-ΔC, as expected (*Figure 3F*). Mutants of Nxf2, which had individual domains removed, uniformly failed to rescue *nxf2* knockdown in OSCs (*Figure 3—figure*

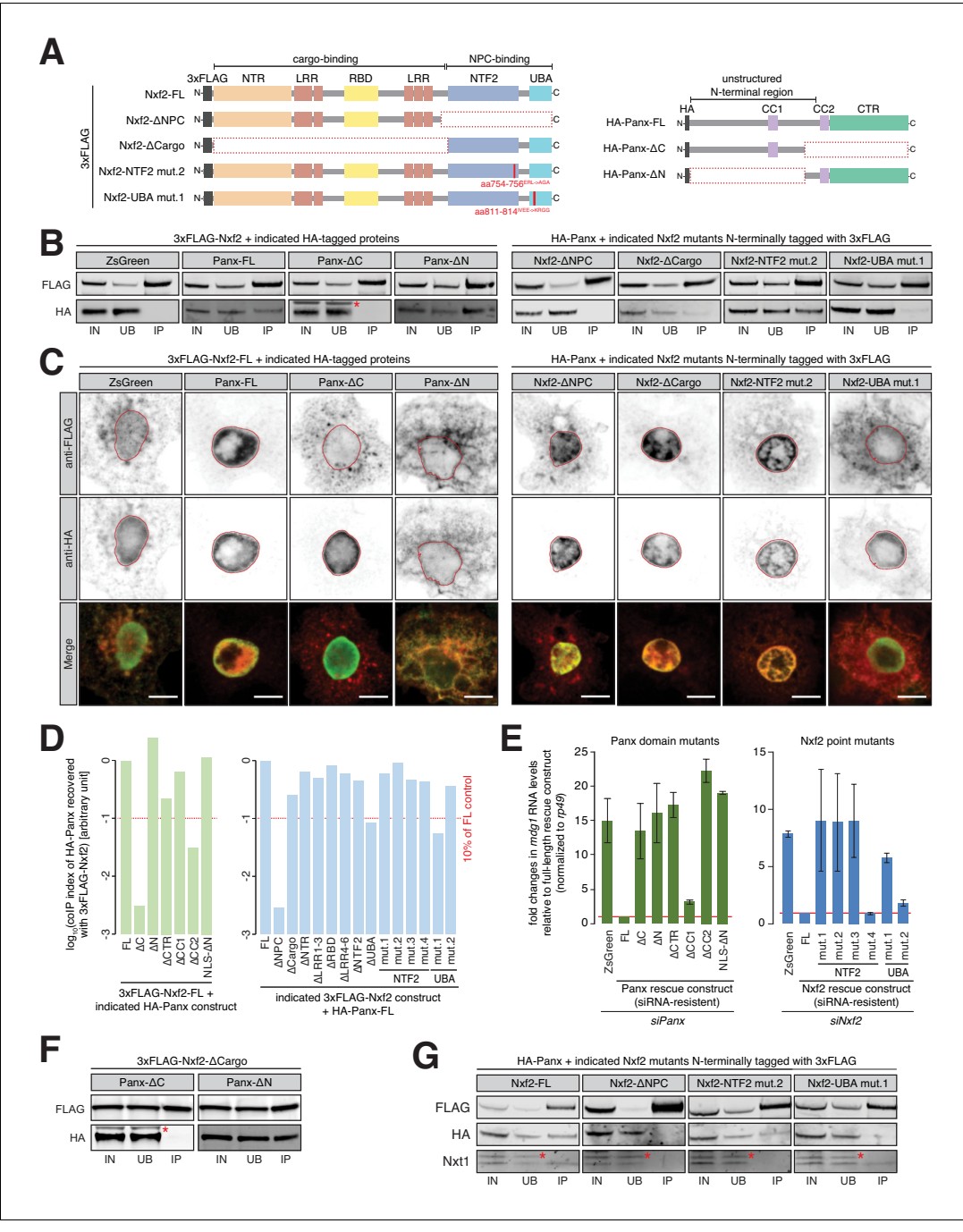

**Figure 3.** Requirements for the formation and function of the PICTS complex. (**A**) Cartoons displaying the Nxf2 and Panx protein structures and domain mutants used (see also *Figure 3—figure supplement 1A* and *Figure 3—figure supplement 2A*). NTR, amino-terminal region; LRR, leucine rich repeats; RBD, RNA-binding domain; NTF2, NTF2-like domain; UBA, Ubiquitin associated domain; CC, coiled coil domain; HA, Hemagglutinin tag. (**B**) Western blot analyses of FLAG-tag co-immunoprecipitation from lysates of S2 cells transfected with the indicated expression constructs (IN, input; UB, unbound; IP, immunoprecipitate). Asterisk indicates unspecific band from anti-HA antibody (see also *Figure 3—figure supplement 1C*, *Figure 3—figure supplement 2B* and *Figure 3—figure supplement 2E*). (**C**) Expression and localization of the indicated Nxf2 (3xFLAG-tagged, top, shown in red in the merge) and Panx (HA-tagged, bottom, shown in green in the merge) expression constructs in transfected S2 cells are shown by immunofluorescence (see also *Figure 3—figure supplement 1D*, *Figure 3—figure supplement 2C* and *Figure 3—figure supplement 2F*). Lamin staining (red lines) was used to draw the outline of the nuclear envelope. Scale bar, 5 µm. (**D**) Bar graphs showing quantification of (**B**) and *Figure 3—figure*

*Figure 3 continued on next page*

*Figure 3 continued*

**supplement 1D** and **Figure 3—figure supplement 2B,E**. Co-immunoprecipitation (coIP) index was calculated from HA[IP/input] over FLAG[IP/input] (see also **Figure 3—figure supplement 1B**). (E) Bar graphs showing fold changes in steady-state RNA levels of the *mdg1* transposon in total RNA from OSCs transfected with siRNAs against Panx (left) and Nxf2 (right) and the indicated siRNA-resistant expression constructs (relative to full-length rescue construct and normalized to *rp49*). Error bars indicate standard deviation (Panx, n = 2; Nxf2, n = 3). (F) As in (B) but showing co-immunoprecipitation of Nxf2-ΔCargo with Panx-ΔC and Panx-ΔN constructs. (G) As in (B) but showing co-immunoprecipitation of HA-Panx and Nxt1 recovered with the indicated Nxf2 expression constructs. Asterisks indicate an unspecific band from anti-Nxt1 antibody.

DOI: https://doi.org/10.7554/eLife.47999.010

The following figure supplements are available for figure 3:

**Figure supplement 1.** Interaction between Nxf2 and Panx is required for proper transposon silencing.
DOI: https://doi.org/10.7554/eLife.47999.011

**Figure supplement 2.** Interaction between Nxf2 and Panx is required for proper transposon silencing.
DOI: https://doi.org/10.7554/eLife.47999.012

---

**supplement 2D**), yet all but Nxf2-ΔUBA still co-purified with full-length Panx (**Figure 3D** and **Figure 3—figure supplement 2B–C**). We also generated point mutants within the UBA domain, altering 2–4 highly conserved amino acids at a time (**Figure 3—figure supplement 2A** right). UBA mutant #1, showed a phenotype similar to the domain deletion with reduced binding to Panx, increased cytoplasmic localization, and failure to rescue loss of Nxf2 (**Figure 3B–E** and **Figure 3—figure supplement 2E–F**). Thus, the interaction of Nxf2 and Panx relies on an intact UBA domain and requires the carboxy-terminal portion of Panx.

The NTF2-like fold was previously shown to mediate the interaction of NXF proteins with Nxt1 (**Herold et al., 2000**; **Kerkow et al., 2012**; **Suyama et al., 2000**). To probe a potential requirement for Nxt1 in silencing, we generated NTF2 domain point mutants in residues involved in the interaction with Nxt1 (**Kerkow et al., 2012**). All four Nxf2-NTF2 point mutants localized predominantly to the nucleus and co-precipitated quantities of Panx comparable to the full-length control (**Figure 3D** and **Figure 3—figure supplement 2E–F**). Yet, three mutants (#1, #2, and #3) failed to rescue transposon de-repression upon depletion of Nxf2 (**Figure 3E**), pointing to an involvement of Nxt1 in silencing. Indeed, reduced amounts of Nxt1 were recovered with the NTF2 mutants #1 and #2, while the NTF2 mutant #4, which rescued transposon expression (**Figure 3E**), as well as UBA mutants #1 and #2 showed levels comparable to full-length Nxf2 (**Figure 3G** and **Figure 3—figure supplement 2G**).

## Tethering of Nxf2 and Nxt1 to RNA triggers silencing

Artificial tethering of Panx to nascent RNA or DNA was previously shown to result in co-transcriptional silencing and the concurrent accumulation of repressive chromatin marks (**Sienski et al., 2015**; **Yu et al., 2015**). To test whether Nxf2 could induce TGS, we created an integrated sensor comprising the *Drosophila simulans* ubiquitin promoter driving an HA-tagged ZsGreen transcript with 9 BoxB sites in its 3' UTR in OSCs (**Figure 4A**). As expected from previous studies (**Sienski et al., 2015**; **Yu et al., 2015**), expression of λN-Piwi did not lead to reduced RNA or protein levels (**Figure 4B** and **Figure 4—figure supplement 1A**), although it did localize to nuclei (**Figure 4—figure supplement 1B**). Tethering of λN-Panx, in contrast, resulted in robust repression of sensor RNA and protein signals, as reported (**Sienski et al., 2015**; **Yu et al., 2015**). λN-Nxf2 caused an even stronger reduction of RNA and protein expression from the reporter (**Figure 4B**). FISH experiments supported consistent repression by Panx and Nxf2 (**Figure 4—figure supplement 1C**). Tethering of λN-Nxt1 also induced reporter repression (**Figure 4B**). Strikingly, upon tethering of Nxt1, Nxf2 or Panx, we observed increased levels of H3K9me3 over the reporter (**Figure 4C**). These data suggest that Nxf2 and Nxt1, along with Panx, act as key effectors of co-transcriptional silencing and are each sufficient to recruit the downstream silencing machinery.

We also tested whether Nxf2 could silence artificial targets upon tethering to DNA rather than nascent transcripts. Our sensor construct carried 8 LacO sites upstream of the *D. sim.* ubiquitin promoter, which drives the expression of HA-ZsGreen (**Figure 4D**). While tethering of LacI-Piwi did not affect sensor expression, LacI-Panx resulted in robust reductions in both RNA and protein levels

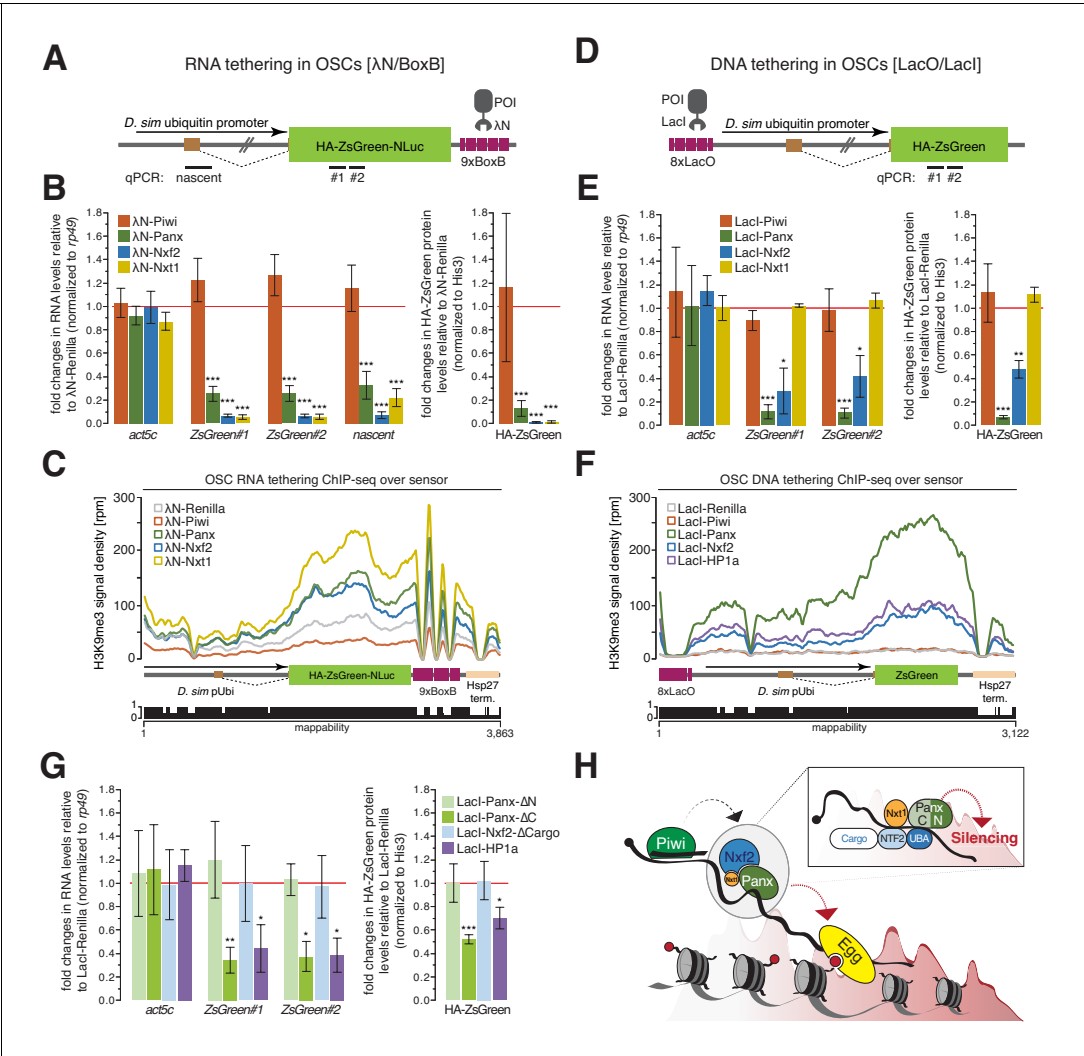

**Figure 4.** Recruitment of PICTS components to nascent RNA results in chromatin silencing. (**A**) Cartoon displaying the RNA tethering sensor construct used in OSCs. The construct was stably integrated and features the *Drosophila simulans* ubiquitin promoter (which contains an intron), the ZsGreen coding sequence (fused to the HA-tag, a nuclear localization sequence and the NLuc luciferase), and a 3' UTR containing 9x BoxB sites. Amplicons for qPCR are indicated. (**B**) Left: Bar graphs showing fold changes in steady-state RNA levels of the sensor and *act5c* in total RNA from OSCs transfected with the indicated λN-fusion expression constructs (relative to a λN-Renilla construct and normalized to *rp49*). Right: Bar graphs showing fold changes in protein levels of HA-ZsGreen in lysates from OSCs transfected with the indicated λN expression constructs (relative to a λN-Renilla construct and normalized to His3). *** denotes P value < 0.0001 (unpaired t-test). Error bars indicate standard deviation (n = 4). (**C**) Density profiles of normalized reads from H3K9me3 ChIP-seq experiments mapping to the tethering reporter as indicated in the cartoon below. The mappability of reads is shown below. (**D**) as in (**A**) but showing the DNA tethering sensor. The construct features 8x LacO binding sites followed by the *Drosophila simulans* ubiquitin promoter and the HA-ZsGreen coding sequence. Amplicons for qPCR are indicated. (**E**) as in (**B**) but showing quantification of DNA tethering experiments. * denotes P value < 0.01; ** denotes p<0.001; *** denotes p<0.0001 (unpaired t-test). Error bars indicate standard deviation (n = 3). (**F**) as in (**C**) but showing density profiles of normalized reads from the DNA tethering reporter. (**G**) Quantification of DNA tethering experiments normalized to *rp49* (qPCR) or His3 (protein) and relative to the same LacI-Renilla construct used in (**E**). * denotes P value < 0.01; ** denotes p<0.001; *** denotes p<0.0001 (unpaired t-test). Error bars indicate standard deviation (n = 3). (**H**) Model for piRNA-guided co-transcriptional silencing. Piwi scans for nascent transposon transcripts and upon target engagement recruits the PICTS complex to the RNA via an unknown signal. The association of the PICTS complex with target transcript causes co-transcriptional silencing, with the amino-terminus of Panx being required for silencing (insert). PICTS complex recruitment results in chromatin remodeling (H3K4me2 removal, H3K9me3 deposition), and depends on general silencing machinery factors such as Egg/dSetDB1.

DOI: https://doi.org/10.7554/eLife.47999.013

The following figure supplement is available for figure 4:

**Figure supplement 1.** Artificial recruitment of Nxf2 to RNA or DNA results in chromatin silencing.

DOI: https://doi.org/10.7554/eLife.47999.014

(*Figure 4E* and *Figure 4—figure supplement 1D*), as previously reported (*Sienski et al., 2015*). LacI-Nxf2 also silenced the reporter, reducing both mRNA and protein output, albeit to a lesser extent than LacI-Panx tethering. Surprisingly, LacI-Nxt1 was unable to silence the sensor construct. Repression by tethered Panx or Nxf2 resulted in a striking decrease in H3K4me2 marks over the transcribed parts (i.e. promoter and ZsGreen coding region), while the remainder of the reporter showed low read coverage and little difference in the prevalence of the mark (*Figure 4—figure supplement 1E*). Conversely, H3K9me3 increased upon repression (*Figure 4F*). Of note, the entire reporter sequence (except a small gap around the LacO site and the 3' UTR where the mappability is poor) was prominently decorated with H3K9me3, suggesting spreading of this chromatin mark following initial silencing.

The data presented above identified functional elements within Nxf2 and Panx that are required for proper localization, interaction, and Piwi-dependent transcriptional gene silencing. We next examined the ability of Panx and Nxf2 mutants to silence our artificial DNA reporters, thus bypassing Piwi-piRNA dependent target recognition. Neither LacI-Nxf2-ΔCargo nor LacI-Panx-ΔN, which were predicted to interact with their full-length partner in OSCs (*Figure 3B–D*), were able to repress the sensor (*Figure 4G*). Yet, LacI-Panx-ΔC, which could not interact with Nxf2, substantially reduced RNA and protein expression from the sensor, with its effects as robust as upon DNA tethering of LacI-HP1a (*Figure 4G*). This suggests that the amino-terminal part of Panx is necessary and sufficient to enforce silencing of an artificial reporter independent of its interaction with Nxf2.

## Discussion

Our data identify Nxf2 and Nxt1 as critical mediators of co-transcriptional gene silencing, acting in concert with Panx to repress loci in response to Piwi-piRNA target engagement (*Figure 4H*). The emerging model for piRNA-dependent silencing implies that target recognition by Piwi is necessary to recruit the PICTS complex onto the appropriate nascent RNA targets. Difficulties in detecting stable interactions between Piwi and PICTS components in vivo may arise from a requirement for Piwi target engagement to licence it for recruitment of silencing complexes, as has been suggested previously (*Sienski et al., 2015*; *Yu et al., 2015*). The same mechanism may underlie the difficulties experienced in observing Piwi on its target loci by ChIP (*Marinov et al., 2015*).

We find that Panx and Nxf2 are interdependent for their protein stability and proper subcellular localization, underscoring the fact that correct assembly of the PICTS complex is essential for TGS, while the silencing capacity, per se, resides in Panx. Of note, previous work reported a partial destabilization of Nxf2 in cells depleted of Nxt1 (*Herold et al., 2001*), potentially extending the interdependency to all three proteins. RIP-seq experiments from unperturbed cells found transposon RNAs enriched only with Panx, as reported (*Sienski et al., 2015*), but not with Nxf2 (*Figure 4—figure supplement 1F*), possibly due to low substrate availability combined with an insensitive assay. These results are consistent with another recent report that did not detect transposon enrichment in Nxf2 CLIP-seq from wild-type cells (*Batki et al., 2019*). However, two other studies identified transposon mRNA association with Nxf2 in CLIP-seq experiments upon depletion of the previously described TGS factor, Mael (*Zhao et al., 2019*), or by using a stable cell line and depletion of endogenous Nxf2 (*Murano et al., 2019*). Considered together, these data suggest that Nxf2 might be important for stabilizing the binding of Panx to nascent RNAs. However, precisely how Nxf2 executes this function remains to be fully elucidated. Of note, Murano and colleagues find that Panx interacts with Nxf2, Piwi, Mael and Arx (*Murano et al., 2019*), which could imply that other TGS factors come into contact with the PICTS complex, although the relationship between these factors and PICTS requires further investigation.

Mutational analyses suggest that Panx and Nxf2 must normally bind Nxt1 to carry out transposon repression. Direct recruitment of any of the PICTS complex components to RNA reporters results in robust chromatin silencing. Upon tethering to DNA, however, Panx induces potent TGS, whereas Nxf2 leads to less prominent effects and Nxt1 shows no silencing capability in our assays. Interestingly, recruitment of the amino-terminal part of Panx alone is necessary and sufficient to induce reporter repression, pinpointing this domain of Panx as harboring the silencing effector function. Future investigations will be crucial to uncover the molecular mechanism by which the Panx amino terminus instructs the downstream chromatin silencing machinery.

Our work, and that of others (*Batki et al., 2019*; *Murano et al., 2019*; *Zhao et al., 2019*) indicates that piRNA-guided co-transcriptional silencing of transposons has coopted several components of the RNA export machinery, namely Nxf2 and Nxt1. Of the four NXF proteins present in flies, only two have thus far been characterized. Interestingly, while Nxf1, acting along with Nxt1, is crucial for canonical mRNA export (*Braun et al., 2001*; *Fribourg et al., 2001*; *Herold et al., 2001*; *Wilkie et al., 2001*), Nxf2 has been coopted by the piRNA pathway and functions in co-transcriptional gene silencing. Nxf3, which also is required for transposon repression in germ cells (*Czech et al., 2013*), is emerging as being critical for the export of piRNA precursors generated from dual-strand clusters in the germline (*ElMaghraby et al., 2019*; *Kneuss et al., 2019*. The role of Nxf4, whose expression is testis-specific, is yet to be established. This remarkable functional diversity of NXF family members correlates with tissue-specific expression patterns, and seems conserved in mammals (*Yang et al., 2001*). However, deciphering how each achieves substrate specificity will be critical to understanding how these homologs can be exclusively dedicated to different targets and confer different fates upon the RNAs that they bind.

Importantly, the fate of the nascent transcript that is detected by Piwi and instructed by PICTS for silencing remains unclear. One hypothesis is that instead of being exported, these targets undergo degradation by the nuclear exosome. Such mechanism would be contrary to yeast, where the TREX complex subunit Mlo3 was shown to oppose siRNA-mediated heterochromatin formation at gene loci (*Yu et al., 2018*), and suggests that different lineages have evolved different silencing mechanisms. In any case, it is possible that a single transcript from a locus that is marked for silencing might pose a lesser threat than an unsilenced locus and, therefore, not be capable of exerting evolutionary pressure for the detemation of its fate.

## Materials and methods

### Fly stocks and handling

All flies were kept at 25˚C. Flies carrying a BAC transgene expressing GFP-Panx were generated by the Brennecke lab (*Handler et al., 2013*). Panx frameshift mutants *panx^M1^* and *panx^M4^* were described earlier (*Yu et al., 2015*). The GFP-Nxf2 fusion knock-in and *nxf2* frameshift mutations (*nxf2[F10\*]* and *nxf2[Δ1\*]*) were generated for this study (see below). Control *w^1118^* flies were a gift from the University of Cambridge Department of Genetics Fly Facility. For knockdowns we used a stock containing the Dcr2 transgene and a nos-GAL4 driver (described in *Czech et al., 2013*) and dsRNA lines from the VDRC (*panx^KK102702^*, *nxf2^KK101676^*, *piwi^KK101658^*). Fertility of the *nxf2* and *panx* mutant females was scored by crossing ten freshly hatched females to five *w^1118^* males and counting the number of eggs laid in 12 hr periods and pupae that developed after 7 days.

### Generation of mutant and transgenic fly strains

Frameshift mutant alleles of *nxf2* were generated by injecting pCFD4 (addgene plasmid # 49411; *Port et al., 2014*) containing two gRNAs against Nxf2 (generated by Gibson assembly) into embryos expressing vas-Cas9 (Bloomington stock 51323). To generate GFP-Nxf2 fusion knock-in flies, homology arms of approximately 1 kb were cloned into pUC19 by Gibson assembly and co-injected with pCFD3 (addgene plasmid # 49410; *Port et al., 2014*) containing a single guide RNA into embryos expressing vas-Cas9 (# 51323, Bloomington stock center). Microinjection and fly stock generation was carried out by the University of Cambridge Department of Genetics Fly Facility. Mutant and knock-in flies were identified by genotyping PCRs and confirmed by sanger sequencing.

### Immunoprecipitation from ovary lysates and mass spectrometry

Ovaries from ~170 GFP-Panx, GFP-Nxf2 and control flies (3–5 days old) were dissected in ice-cold PBS and lysed in 300 µl of CoIP Lysis Buffer (20 mM Tris-HCl pH 7.5, 150 mM NaCl, 2 mM MgCl2, 10% glycerol, 1 mM DTT, 0.1 mM PMSF, 0.2% NP-40 supplemented with complete protease inhibitors [Roche]) and homogenized using a motorized pestle. Lysates were cleared for 5 min at 16000 g and the residual pellet re-extracted with the same procedure. GFP-tagged proteins were immunoprecipitated by incubation with 30 µl of GFP-Trap magnetic beads (Chromotek) for 3 hr at 4˚C on a tube rotator. The beads were washed 6x with Lysis Buffer and 2x with 100 mM Ammonium

Bicarbonate, before TMT-labelling followed by quantitative Mass Spectrometry. TMT chemical iso-baric labeling were performed as described (*Papachristou et al., 2018*).

## Analysis of mass spectrometry data

Raw data were processed in Proteome Discoverer 2.1 software (Thermo Fisher Scientific) using the SequestHT search engine. The data were searched against a database derived from FlyBase ('*dmel-all-translation-r6.24*') at a 1% spectrum level FDR criteria using Percolator (University of Washington). For the SequestHT node the following parameters were included: Precursor mass tolerance 20 ppm and fragment mass tolerance 0.5 Da. Dynamic modifications were oxidation of M (+15.995 Da), dea-midation of N, Q (+0.984 Da) and static modifications were TMT6plex at any N-Terminus and K (+229.163 Da). The consensus workflow included S/N calculation for TMT intensities and only unique peptides identified with high confidence (FDR < 0.01) were considered for quantification. Down-stream data analysis was performed on R using the qPLEXanalyzer package (https://doi.org/10.5281/zenodo.1237825) as described (*Papachristou et al., 2018*). Only proteins with more than one unique peptide were considered.

## Cell culture

*Drosophila* Ovarian Somatic Cells (OSCs) were a gift from Mikiko Siomi and were cultured at 26°C in Shields and Sang M3 Insect Medium (Sigma Aldrich) supplemented with 0.6 mg/ml Glutathione, 10% FBS, 10 mU/ml insulin and 10% fly extract (purchased from DGRC) as described (*Niki et al., 2006*; *Saito, 2014*; *Saito et al., 2009*). Cell identity was authenticated by whole genome DNA sequencing in-house. Gibco *Drosophila* Schneider 2 (S2) cells were purchased from Thermo Fisher Scientific (catalog number R69007) and were grown at 26°C in Schneider's *Drosophila* Media (Gibco) supplemented with 10% heat-inactivated FBS. Cell identity was characterized by Thermo Fisher Sci-entific through isozyme and karyotype analysis (see product description). OSCs and S2 cells tested negative for mycoplasma contamination in-house. Knockdowns (all siRNA sequences are given in *Supplementary file 1*) and transfections in OSCs were carried out as previously described (*Saito, 2014*). In short, for knockdown experiments 10 × 10⁶ cells were nucleofected with 200 pmol annealed siRNAs using the Amaxa Cell Line Nucleofector Kit V (Lonza, program T-029). After 48 hr, 10 × 10⁶ cells were nucleofected again with 200 pmol of the same siRNA and allowed to grow for an additional 48 hr before further processing. For rescue experiments, 5 µg of rescue construct plas-mid were added to the second knockdown solution. OSCs were transfected with 10 µg of plasmid using Xfect (Clontech), according to manufacturer's instruction. S2 cells were transfected with 2 µg of plasmid using Effectene (Qiagen), according to manufacturer's instructions.

## Co-immunoprecipitation from cell lysates

S2 cells or OSCs were transfected with 3xFLAG- and HA-tagged constructs (wild-type and mutants). Cells were harvested 48 hr after transfection and lysed in 250 µl of CoIP Lysis Buffer (Pierce) supple-mented with Complete protease inhibitors (Roche). 200 µg of proteins for each sample were diluted to 1 ml with CoIP Lysis Buffer and the 3xFLAG-tagged bait was immunoprecipitated by incubation with 20 µl of anti-FLAG M2 Magnetic Beads (Sigma M8823) for 2 hr at 4°C on a tube rotator. The beads were washed 3 × 15 min with TBS supplemented with protease inhibitors. Beads were then resuspended in 2x NuPAGE LDS Sample Buffer (Thermo Fisher Scientific) without reducing agent and boiled for 3 min at 90°C to elute immunoprecipitated proteins. IPs, unbound fractions and input fractions were diluted to 1x NuPAGE LDS Sample Buffer concentration and reducing agent was added. Samples were boiled at 90°C for 10 min before separating proteins as described below.

## Western blot

Protein concentration was measured using a Direct Detect Infrared Spectrometer (Merck). 20 µg of proteins were separated on a NuPAGE 4–12% Bis-Tris gel (Thermo Fisher Scientific). Proteins were transferred with an iBLot2 device (Invitrogen) on a nitrocellulose membrane and blocked for 1 hr in 1x Licor TBS Blocking Buffer (Licor). Primary antibodies were incubated over night at 4°C. Licor sec-ondary antibodies were incubated for 45 min at room temperature (RT) and images acquired with an Odyssey CLx scanner (Licor). The following antibodies were used: anti-HA (ab9110), anti-FLAG (Sigma #F1804), anti-GFP (ab13970), anti-Piwi (described in *Brennecke et al., 2007*), anti-Nxt1

(*Herold et al., 2001*), anti-Histone H3 (ab10799), anti-Tubulin (ab18251), mouse anti-Panx (*Sienski et al., 2015*), IRDye 680RD Donkey anti-Rabbit IgG (H + L) (Licor), IRDye 800CW Donkey anti-Mouse IgG (H + L) (Licor), IRDye 800CW Goat anti-Rat IgG (H + L) (Licor).

## Immunofluorescence in ovaries

Fly ovaries were dissected in ice-cold PBS and fixed in 4% paraformaldehyde (PFA) at RT for 15 min. After two quick rinses in PBS with Triton at 0.3% (PBS-Tr), samples were permeabilized with 3 × 10 min washes with PBS-Tr. Samples were then blocked in PBS-Tr with 1% BSA for 2 hr at RT and then incubated overnight at 4°C with primary antibodies in PBS-Tr and 1% BSA. The next day, samples were washed 3 × 10 min at RT in PBS-Tr and incubated overnight at 4°C with secondary antibodies in PBS-Tr and 1% BSA. The next day, samples were washed 4 × 10 min in PBS-Tr at RT and DAPI (Thermo Fisher Scientific #D1306) was added during the third wash. After 2 × 5 min washes in PBS, samples were mounted on slides with ProLong Diamond Antifade Mountant (Thermo Fisher Scientific #P36961) and imaged on a Leica SP8 confocal microscope (63x and 100x Oil objective). The following antibodies were used: chicken anti-GFP (ab13970), rabbit anti-Piwi (described in *Brennecke et al., 2007*), mouse anti-Aub (*Senti et al., 2015*), anti-Rabbit-555 (Thermo Fisher), anti-Mouse-647 (Thermo Fisher), anti-Chicken-647 (Abcam).

## Immunofluorescence from cells

Cells were plated one day in advance on Fibronectin- or Concanavalin A- coated coverslips (for OSCs and S2 cells, respectively), fixed for 15 min in 4% PFA, permeabilized for 10 min in PBS with 0.2% Triton (PBST) and blocked for 30 min in PBS, 0.1% Tween-20% and 1% BSA. Primary antibodies were diluted in PBS, 0.1% Tween-20% and 0.1% BSA and incubated overnight at 4°C. After 3 × 5 min washes in PBST, secondary antibodies were incubated for 1 hr at RT. After 3 × 5 min washes in PBST, DAPI was incubated for 10 min at RT, washed two times and the coverslips were mounted using ProLong Diamond Antifade Mountant (Thermo Fisher Scientific #P36961) and imaged on a Leica SP8 confocal microscope (100x Oil objective). The following antibodies were used: anti-Lamin (Developmental Studies Hybridoma Bank ADL67.10), anti-HA (ab9111), anti-FLAG (Cell Signaling Technology 14793S), anti-chicken-488 (Abcam), anti-Rabbit-555 (Thermo Fisher), anti-Mouse-647 (Thermo Fisher).

## RNA fluorescent in situ hybridization (RNA FISH)

RNA FISH was performed with Hybridization Chain Reaction (HCR), similar as reported (*Ang and Yung, 2016*; *Choi et al., 2014*). OSCs were fixed for 15 min in 4% PFA, washed 2 × 5 min with PBS and permeabilized for at least 24 hr in 70% ethanol at −20°C. Ethanol was removed and slides washed twice for 5 min in 2x Saline-Sodium Citrate buffer (SSC). Priming for hybridization was done by incubating for 10 min in 15% formamide in 2x SSC. HCR probes were diluted to 1 nM each in hybridization buffer (15% formamide, 10% dextran sulfate in 2x SSC) and incubated overnight at 37°C in a humidified chamber. Excess probes were removed by rinsing twice in 2x SSC and washing once in 30% formamide for 10 min at 37°C. HCR hairpins conjugated to AlexaFluor-647 (IDT) were heat-denatured and diluted to 120 nM in 5x SSC and 0.1% Tween-20. HCR amplification was carried out for 2 hr at RT in the dark and washed 3 × 10 min with 5x SSC and 0.1% Tween-20. Nuclei were stained with DAPI for 10 min, followed by 3 × 10 min washes in 5x SSC. Slides were mounted with ProLong Diamond Antifade Mountant (Thermo Fisher Scientific) and imaged on a Leica SP8 confocal microscope (100x Oil objective). The sequences of all probes are given in *Supplementary file 1*.

## Image analysis

Intensity plot profiles across individual egg chambers were acquired in Fiji (lines displayed). Intensity values for each channel were averaged over 10 pixels and adjusted as a percentage of the highest value along the profile. A threshold of 30% DAPI intensity was set to define nuclei. Individual egg chambers used for analysis are displayed for each channel with inverted LUT.

## Tethering experiments

For RNA tethering, OSCs with a stable integration of the sensor plasmid (pDsimUbi-HA-ZsGreen-NLuc-9xBoxB) were generated in the lab. $4 \times 10^6$ cells were nucleofected with 5 µg of plasmid

expressing λN-3xFLAG-tagged constructs, as described above. After 48 hr, $4 \times 10^6$ cells were nucleofected again with 5 µg of the same plasmid and allowed to grow for an additional 48 hr before the relative expression of the sensor was analyzed. For DNA tethering, OSCs were transiently transfected with 8xLacO-pDsimUbi-HA-ZsGreen sensor plasmid and LacI-3xFLAG fusion constructs. Cells were allowed to grow for 72 hr before the relative expression of the sensor was determined.

## ChIP-seq from ovaries

Ovaries from 120 to 150 adult flies were dissected in ice-cold PBS, collected in 1.5 ml Bioruptor Microtubes (Diagenode #C30010016), and immediately frozen at −80˚C. Samples were crosslinked in 1 ml A1 buffer (60 mM KCl, 15 mM NaCl, 15 mM HEPES pH 7.6, 4 mM MgCl2, 0.5% Triton X-100, 0.5 mM dithiothreitol (DTT), 10 mM sodium butyrate and complete EDTA-free protease inhibitor cocktail [Roche #04693159001]), in the presence of 1.8% formaldehyde. Samples were homogenized with a micropestle for 2 min and incubated for a total time of 15 min at RT on a rotating wheel. Crosslinking was stopped by adding 225 mM glycine followed by incubation for 5 min on a rotating wheel. The homogenate was centrifuged for 5 min at 4,000 g at 4˚C. The supernatant was discarded, and the nuclear pellet was washed twice in 1 ml A1 buffer and once in 1 ml of A2 buffer (140 mM NaCl, 15 mM HEPES pH 7.6, 1 mM EDTA, 0.5 mM EGTA, 1% Triton X-100, 0.5 mM DTT, 0.1% sodium deoxycholate, 10 mM sodium butyrate and complete mini EDTA-free protease inhibitor cocktail) at 4˚C. Nuclei were then resuspended in 100 µl A2 buffer with 1% SDS and 0.5% N-laurosyl-sarcosine and incubated for 2 hr at 4˚C with agitation at 1,500 rpm. Chromatin was sonicated using a Bioruptor Pico (Diagenode #B01060010) for 16 cycles of 30 s on/30 s off. Sheared chromatin size peaked at 150 bp. After sonication and 5 min high-speed centrifugation at 4˚C, fragmented chromatin was recovered in the supernatant and the final volume was raised to 1 ml in A2 buffer with 0.1% SDS. 50 µl of the diluted samples were used as DNA input control, in a final volume of 200 µl of A2 buffer with 0.1% SDS. Chromatin for IP was precleared by addition of 15 µl of Protein A/G Magnetic Beads (Thermo Fisher Scientific) suspension followed by overnight incubation at 4˚C. Beads were removed by centrifugation, and anti-H3K9me3 (Active Motif #39161) antibody was added (1:200 dilution) to 5 µg of chromatin and incubated for 4 hr at 4˚C on a rotating wheel. 50 µl of Protein A/G Magnetic Beads were added, and incubation was continued overnight at 4˚C. Antibody-protein complexes were washed 4 times in A3 (A2 +0.05% SDS) buffer and twice in 1 mM EDTA, 10 mM Tris (pH 8) buffer for 5 min at 4˚C on a rotating wheel. Chromatin was eluted from the beads in 200 µl of 10 mM EDTA, 1% SDS, 50 mM Tris (pH 8) for 30 min with agitation at 1,500 rpm and then reverse-crosslinked overnight at 65˚C, together with the input DNA. IP and input samples were treated with 2 µl of Proteinase K (Thermo Fisher Scientific #EO0491) for 3 hr at 56˚C. DNA was purified using the MinElute PCR purification Kit (Thermo Fisher Scientific), according to manufacturer's instructions, and resuspended in 30 µl water. Recovered DNA was quantified with Qubit 4 Fluorometer (Thermo Fisher Scientific) and analysed with Agilent Bioanalyzer 2100 High Sensitivity DNA Chip (Agilent). DNA libraries were prepared with NEBNext Ultra II DNA Library Prep Kit for Illumina (NEB), according to manufacturer's instructions. DNA libraries were quantified with KAPA Library Quantification Kit for Illumina (Kapa Biosystems) and deep-sequenced with Illumina HiSeq 4000 (Illumina).

## ChIP-seq from OSCs

For ChIP from OSCs we adapted a protocol by Schmidt and colleagues (*Schmidt et al., 2009*). In short, $10 \times 10^6$ OSCs were crosslinked in 1% formaldehyde for 10 min. Crosslinking was quenched by addition of glycine solution, followed by three washes in ice-cold PBS. Crosslinked cells were either snap-frozen in liquid nitrogen and stored at −80˚C or processed immediately. Cells were resuspended in 1 ml buffer LB1 (50 mM HEPES-KOH pH 7.5, 140 mM NaCl, 1 mM EDTA, 10% glycerol, 0.5% Igepal CA-630, 0.25% Triton-X 100, EDTA-free protease inhibitor cocktail [Roche]) and incubated on ice for 10 min while inverting several times. Cells were centrifuged for 5 min at 2,000 g at 4˚C. Supernatant was discarded and pellet resuspended in 1 ml LB2 10 mM Tris-HCL pH 8.0, 200 mM NaCl, 1 mM EDTA, 0.5 M EGTA, EDTA-free protease inhibitor cocktail). Cells were incubated on ice for 5 min and centrifuged again. Isolated nuclei were resuspended in 300 µl sonication buffer LB3 (10 mM Tris-HCL pH 8, 100 mM NaCl, 1 mM EDTA, 0.5 mM EGTA, 0.1% Na-Deoxycholate, 0.5% N-lauroylsarcosine, EDTA-free protease inhibitor cocktail) and transferred in 1.5 ml Bioruptor Microtubes (Diagenode). Chromatin was sonicated using a Bioruptor Pico (Diagenode) for 16 cycles

of 30 s on/30 s off. Sheared chromatin size peaked at 150 bp. The lysate was cleared by high-speed centrifugation at 4°C. 100 µl Protein A Dynabeads (Thermo Fisher Scientific) were incubated with 5 µl H3K9me3 (Active Motif #39161) or H3K4me2 antibody (Millipore # 07–030) over night at 4°C while rotating. The cleared lysate was split in two equal fractions and a 5 µl input fraction was saved for further processing. Lysate volumes were adjusted to 300 µl with LB3 and Triton-X 100 was added to a final concentration of 1%. Lysates were incubated with either H3K9me3 or H3K4me2 coated beads over night at 4°C while rotating. Washing, reverse-crosslinking, DNA purification and library preparation was done as described above (ChIP-seq from ovaries).

## RNA isolation

Cell pellets or fly ovaries were lysed in 1 ml Trizol and RNA was extracted using RNeasy mini prep column (Qiagen), according to manufacturer's instructions.

## qPCR analysis

1 µg of total RNA was treated with DNAseI (Thermo Fisher Scientific), according to manufacturer's instructions. Reverse transcription was performed with Superscript III First Strand Synthesis Kit (Thermo Fisher Scientific), using oligo(dT)$_{20}$ primers, according to the manufacturer's instructions. Real-time PCR (qPCR) experiments were performed with a QuantStudio Real-Time PCR Light Cycler (Thermo Fisher Scientific). Transposon levels were quantified using the ∆∆CT method (*Livak and Schmittgen, 2001*), normalized to *rp49* and fold changes were calculated relative to the indicated controls. All oligonucleotide sequences are given in *Supplementary file 1*.

## RIP-seq from ovaries

Ovaries from ~100 GFP-Panx or GFP-Nxf2 flies (3–5 days old) were dissected in ice-cold PBS and fixed with 0.1% PFA for 20 min, followed by quenching with equal volumes of 125 mM Glycine. Fixed ovaries were lysed in 200 µl of RIPA Buffer (supplemented with complete protease inhibitors (Roche) and RNasin Plus 40 U/ml) and homogenized using a motorized pestle. Lysates were incubated 3 min at 37°C with 4 µl of Turbo DNase, incubated 20 min at 4°C on a tube rotator and sonicated with a Bioruptor Pico (3 cycles of 30 s on/30 s off). Lysates were pre-cleared using 40 µl of Pierce Protein A/G beads for 1 hr at 4°C and GFP-tagged proteins were immunoprecipitated by incubation with 50 µl of GFP-Trap magnetic agarose beads (Chromotek) overnight at 4°C. An aliquot of pre-cleared input lysate was saved for RNA isolation and library preparation. Following 3 washes in 150 mM KCl, 25 mM Tris (pH 7.5), 5 mM EDTA, 0.5% NP40, 0.5 mM DTT (supplemented with protease inhibitors and RNasin Plus 1:1000), IP and input samples were reverse crosslinked in 1x Reverse Crosslinking buffer (PBS, 2% Nlauroyl sarcosine, 10 mM EDTA, 5 mM DTT) and Proteinase K. RNA isolation was performed using Trizol and 100 ng of input or IP RNA were used for library preparation using the SMARTer stranded RNA-seq Kit (Clontech). DNA libraries were quantified with KAPA Library Quantification Kit for Illumina (Kapa Biosystems) and deep-sequenced with Illumina HiSeq 4000 (Illumina).

## Small RNA-seq library preparation

Small RNA libraries were generated as described previously (*Jayaprakash et al., 2011*). Briefly, 18- to 29-nt-long small RNAs were purified by PAGE from 10 µg of total ovarian RNA. Next, the 3' adapter (containing four random nucleotides at the 5' end) was ligated overnight using T4 RNA ligase 2, truncated KQ (NEB). Following recovery of the products by PAGE purification, the 5' adapter (containing four random nucleotides at the 3' end) was ligated to the small RNAs using T4 RNA ligase (Abcam) for 1 hr. Small RNAs containing both adapters were recovered by PAGE purification, reverse transcribed and PCR amplified prior quantification using the Library Quantification Kit for Illumina (Kapa Biosystems) and sequenced on an Illumina HiSeq 4000 (Illumina). All adapter sequences are given in *Supplementary file 1*.

## RNA-seq library preparation

1 µg of total RNA was used as input material for library preparation. The NEBNext Poly(A) mRNA magnetic Isolation Module (NEB) was used to isolate poly(A) RNAs. Libraries were generated with the NEBNext Ultra Directional RNA Library Prep kit for Illumina (NEB) according to manufacturer's

instructions. The pooled libraries were quantified with KAPA Library Quantification Kit for Illumina (Kapa Biosystems) and sequenced on an Illumina HiSeq 4000 (Illumina).

## RNA-seq, small RNA-seq, RIP-seq and ChIP-seq analysis

Raw fastq files generated by Illumina sequencing were analysed by a pipeline developed in-house. In short, the first and last base of each 50 bp read were removed using fastx trimmer (http://hannon-lab.cshl.edu/fastx_toolkit/). RIP-seq reads were first aligned against rRNAs and mapped reads discarded. High-quality reads were aligned to the *Drosophila melanogaster* genome release 6 (dm6) downloaded from Flybase using STAR (*Dobin et al., 2013*). For transposon-wide analysis, genome multi-mapping reads were randomly assigned to one location using option '–outFilterMultimapNmax 1000 –outMultimapperOrder Random' and non-mapping reads were removed. Alignment files were then converted back to fastq format with samtools (*Li et al., 2009*) and re-aligned to the transposon consensus sequences allowing multi-mappers that were assigned to a random position. Generated bam alignment files were indexed using samtools index. For genome-wide analyses, multi-mapping reads were removed to ensure unique locations of reads. Normalization was achieved by calculating rpm (reads per million) using the deepTools2 bamCoverage function (*Ramírez et al., 2016*) with 10 bp bin sizes. The scaling factor for transposon mapping reads was calculated from reads that aligned to transposon consensus sequences relative to genome aligned reads. Reads mapping to genes were counted with htseq (*Anders et al., 2015*) and transposon derived reads were calculated using a custom script (available with this article as *Source code 1*). Metaplots flanking euchromatic insertion sites and transposon coverage plots were calculated by deepTools2 with bin sizes of 10 bp and 50 bp, respectively. Stranded RNA libraries were trimmed, aligned and indexed as described above. Alignment files were split in sense and antisense reads using samtools view. Normalization of the split alignment files as well as feature counting was performed as described above. For transposon expression analysis only sense reads were considered. Differential expression analysis was performed using a custom build R script (available with this article as *Source code 2*). Adapters from raw small RNA fastq files were clipped with fastx_clipper (adapter sequence AGATCGGAAGAGCACACGTC TGAACTCCAGTCA) keeping only reads with at least 23 bp length. Then the first and last four bases were trimmed using seqtk (https://github.com/lh3/seqtk). Alignment and normalization were performed as described above. Only high-quality small RNA reads with a length between 23 and 29 bp were used for further analysis of piRNA profiles. piRNA distribution was calculated and plotted in R. For piRNA coverage plots over TEs, only the 5' position of reads was plotted.

## Generation of annotation files for RNA-seq and ChIP-seq analysis

The locations of euchromatic transposon insertions in OSCs were derived from *Sienski et al. (2012)* and updated to dm6 genome release coordinates using the UCSC liftOver tool. Transposon consensus sequences were downloaded from Flybase. Mappability tracks for dm6 with 50 bp resolution were calculated as described (*Derrien et al., 2012*). Piwi-dependent OSC insertions were defined by comparing H3K9me3 signal intensities of siRNA-mediated knockdowns for *gfp* and *piwi*. Signal was counted by htseq using a customized GTF file including the locations of all euchromatic TE insertions in OSCs and reads were normalized to rpm. TE insertions were annotated as Piwi-dependent if the ratio of normalized signal intensity of GFP knockdown versus Piwi knockdown was higher than 2.

## Plotting and data visualization

Random genomic windows for box plots of H3K9me3 ChIP-seq data were calculated using BEDtools' random function (*Quinlan and Hall, 2010*) with bin size 5,000 bp, bin number 1000 and random seed number 800. 100 random windows were chosen (number 200–300) and analysed along with ChIP-seq data for de-repressed TEs and those not affected. Welch two sample t-test was applied for statistics. Metaplots of euchromatic TE insertions as well as TE coverage plots for RNA-seq and ChIP-seq data were generated with deepTools2 and Adobe Illustrator. Scatterplots for differentially expressed transposons and genes were generated with R package ggplot2. Heatmaps were calculated with deepTools2 and data plotted in R. For scatterplots, only TEs and genes with a scaled read count larger than 1 (rpm >1) were used in the analysis and included in plots.

## Quantification and statistical analysis

Statistical analysis applied to qPCR data sets was calculated by unpaired (two sample) t Test. The number of biological replicates is indicated in the figure legends. Statistical analysis applied to data sets displayed as box plots (*Figure 1—figure supplement 2A*) was calculated by Welch two sample t-test.

## Data availability

Sequencing data reported in this paper has been deposited in Gene Expression Omnibus under ID code GSE121661. Mass Spectrometry data has been deposited to PRIDE Archive under ID code PXD011415.

## Acknowledgements

We thank the CRUK Cambridge Institute Bioinformatics, Genomics, Microscopy, Proteomics and Research Instrumentation & Cell Services Core Facilities for technical support. We thank Aled Perry for his help in establishing a ChIP protocol for OSCs. We thank Karan Mehta for help with image analysis. We thank the University of Cambridge Department of Genetics Fly Facility for microinjection services and fly stock generation. We thank the Vienna Drosophila Resource Center and the Bloomington Stock Center for fly stocks, J Brennecke for anti-Aub and anti-Panx antibodies, and E Izaurralde for anti-Nxt1 antibody. Research in the Hannon laboratory is supported by Cancer Research UK and by Wellcome Trust award 110161/Z/15/Z. FC is supported by an EMBO Long-Term fellowship. MM is supported by a Boehringer Ingelheim Fonds PhD fellowship.

## Additional information

### Funding

| Funder | Grant reference number | Author |
| --- | --- | --- |
| Wellcome | Investigator award 110161/Z/15/Z | Gregory J Hannon |
| Cancer Research UK | | Gregory J Hannon |
| European Molecular Biology Organization | Long-Term Fellowship ALTF 1015-2017 | Filippo Ciabrelli |
| Boehringer Ingelheim Fonds | PhD fellowship | Marzia Munafò |

The funders had no role in study design, data collection and interpretation, or the decision to submit the work for publication.

### Author contributions

Martin H Fabry, Data curation, Software, Formal analysis, Validation, Investigation, Visualization, Methodology, Writing—original draft; Filippo Ciabrelli, Marzia Munafò, Data curation, Formal analysis, Funding acquisition, Validation, Investigation, Visualization, Methodology, Writing—original draft; Evelyn L Eastwood, Investigation, Visualization, Methodology, Writing—original draft; Emma Kneuss, Federica A Falconio, Resources, Investigation; Ilaria Falciatori, Investigation, Methodology; Gregory J Hannon, Conceptualization, Supervision, Funding acquisition, Project administration, Writing—review and editing; Benjamin Czech, Conceptualization, Data curation, Formal analysis, Supervision, Validation, Investigation, Visualization, Methodology, Writing—original draft, Project administration, Writing—review and editing

### Author ORCIDs

Martin H Fabry https://orcid.org/0000-0002-8484-4715
Filippo Ciabrelli https://orcid.org/0000-0001-7064-4723
Marzia Munafò https://orcid.org/0000-0002-2689-8432
Emma Kneuss http://orcid.org/0000-0003-0662-8539

Gregory J Hannon  https://orcid.org/0000-0003-4021-3898
Benjamin Czech  https://orcid.org/0000-0001-8471-0007

**Decision letter and Author response**
Decision letter https://doi.org/10.7554/eLife.47999.024
Author response https://doi.org/10.7554/eLife.47999.025

## Additional files

### Supplementary files

• Source code 1. Bash script for counting transposon derived reads.
DOI: https://doi.org/10.7554/eLife.47999.015

• Source code 2. R code for differential expression analysis.
DOI: https://doi.org/10.7554/eLife.47999.016

• Supplementary file 1. Oligonucleotides used in this study.
DOI: https://doi.org/10.7554/eLife.47999.017

• Transparent reporting form
DOI: https://doi.org/10.7554/eLife.47999.018

### Data availability

Sequencing data reported in this paper has been deposited in GEO under accession number GSE121661. Mass Spectrometry data has been deposited to the PRIDE Archive (accession number PXD011415).

The following datasets were generated:

| Author(s) | Year | Dataset title | Dataset URL | Database and Identifier |
|---|---|---|---|---|
| Fabry MH, Ciabrelli F, Munafò M, Eastwood EL, Kneuss E, Falciatori I, Falconio FA, Hannon GJ, Czech B | 2019 | piRNA-guided co-transcriptional silencing coopts nuclear export factors | https://www.ncbi.nlm.nih.gov/geo/query/acc.cgi?acc=GSE121661 | NCBI Gene Expression Omnibus, GSE121661 |
| Fabry MH, Ciabrelli F, Munafò M, Eastwood EL, Kneuss E, Falciatori I, Falconio FA, Hannon GJ, Czech B | 2019 | piRNA-guided co-transcriptional silencing coopts nuclear export factors | https://www.ebi.ac.uk/pride/archive/projects/PXD011415 | PRIDE Archive, PXD011415 |

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
