## [Decision Letter]

Thank you for submitting your article "piRNA-guided co-transcriptional silencing coopts nuclear export factors" for consideration by *eLife*. Your article has been reviewed by two peer reviewers, and the evaluation has been overseen by a Reviewing Editor and James Manley as the Senior Editor. The reviewers have opted to remain anonymous.

The reviewers have discussed the reviews with one another and the Reviewing Editor has drafted this decision to help you prepare a revised submission.

The reviewers acknowledge your discovery of a new factor (Nxf2) that is an effector of transposon silencing: piRNA biogenesis is not affected, but silencing of transposons that depend on the piRNA pathway is affected. Moreover, Nxf2 (in complex with Nxt1) interacts with a previously reported effector Panx. Finally, assuming that somehow the Piwi complex recruits Panx and Nxf2 to the target locus, some evidence is proposed to show that the N-term of Panx is needed to recruit the silencing machinery, while the C-term of Panx interacts with Nxf2, which is proposed to be holding on to the nascent RNA. Hence, Panx alone is not sufficient for silencing – Nxf2 is needed too.

The reviewers also raise some questions:

1) What is the direct link between the piRNA-Piwi complex and Panx, which was expected from your previous work (Yu et al., 2015) and from another study from the Brennecke lab (Sienski et al., 2015)? This does not appear valid anymore. So is this at all piRNA-guided?

2) How does the complex (Panx-Nxf2) recruit H3K9me3?

3) Discrepancies are noted with 3 other papers deposited on bioRxiv.

Despite these outstanding questions, the reviewers are prepared to move on with your paper, provided you:

1) Clearly state and discuss the discrepancies between your present and previous findings.

2) Thoroughly discuss the relevant bioRxiv papers dealing with Nxf2, including that there are other interpretations possible.

3) Discuss other known effectors such as Maelstrom, GTSF1 etc. and put this in the context of what we already know.

4) Discuss the relationship between your results and those in *S. pombe*, showing that the nuclear mRNA export factor Mlo3 (human ALY/REF) opposes siRNA-mediated heterochromatin formation at mRNA genes (Yu et al., 2018). Your findings suggest that flies may have evolved a different mechanism that prevents piRNA targets/transposons, which act as templates for recruitment of downstream factors, from being exported.

---

## [Author Response]

[…] 1) What is the direct link between the piRNA-Piwi complex and Panx, which was expected from your previous work (Yu et al., 2015) and from another study from the Brennecke lab (Sienski et al., 2015)? This does not appear valid anymore. So is this at all piRNA-guided?2) How does the complex (Panx-Nxf2) recruit H3K9me3?3) Discrepancies are noted with 3 other papers deposited on bioRxiv.Despite these outstanding questions, the reviewers are prepared to move on with your paper, provided you:1) Clearly state and discuss the discrepancies between your present and previous findings.

Our previous study reported the presence of Piwi in GFP- or 3xFLAG-Panx IPs from OSS cells (Yu et al., 2015). Similar results were obtained by the Brennecke lab using antibodies against endogenous Piwi and Panx from nuclear OSC lysates (Sienski et al., 2015). Indeed, we, as well as Batki et al., were unable to detect significant enrichment of Piwi in either Panx or Nxf2 IP-mass spectrometry experiments (Batki et al: from nuclear OSC lysates; our study: total lysates from ovaries), though the Siomi group did see Piwi in their mass spectrometry of bands selected based upon their size from a Panx immunoprecipitate (Murano et al., 2019, Figure 1A).

We believe that these outcomes are likely due to the interaction between Piwi and downstream factors being transient, only occurring when Piwi engages a target RNA from a de-repressed transposon. It has been proposed by many in the field that target engagement licenses Piwi for recruitment of a highly potent silencing machinery. According to this prevalent model, and bearing in mind that several of these factors independently induce silencing when tethered artificially to the RNA, one might expect that only a small fraction of Piwi would interact with Panx or Nxf2 under normal circumstances.

We certainly were aware that Batki and colleagues find binding of Piwi to Panx in coIP experiments. However, this interaction becomes visible upon analysis of a substantially greater amount of protein than used for their other IPs (400X, see Figure 1B in Batki et al., 2019).

Upon careful examination of our respective coIP experiment reported in this manuscript (Figure 1—figure supplement 1D), we were indeed able to detect a faint band corresponding to Piwi in co-immunoprecipitates with 3xFLAG-tagged Nxf2, Panx, and Nxt1 but not with ZsGreen. This band is more obvious with increased contrast (see updated Figure 1—figure supplement 1D), thus our results are indeed consistent with our previous studies and those of others, and with at least the hypothesis that it is target-engaged Piwi with which these proteins form a complex. We have updated the original image and changed the corresponding paragraphs in the Results and Discussion sections of the manuscript accordingly.

2) Thoroughly discuss the relevant bioRxiv papers dealing with Nxf2, including that there are other interpretations possible.

We have added a paragraph to the Discussion of our manuscript highlighting relevant findings in the other *bioRxiv* papers.

One of the seemingly contradictory results is the RNA-binding activity of Nxf2. Despite considerable efforts, we were not able to identify robust binding of transposon transcripts to Nxf2 in a wild-type context (Figure 4—figure supplement 1F). Similarly, the updated manuscript version (see *bioRxiv* updates) from Batki and colleagues does not report any RIP- or CLIP-seq data for Nxf2, instead only shows the ability of Nxf2’s first unit to bind RNA in vitro.

Zhao et al. find that Nxf2 binds to transposon transcripts in Nxf2-HALO knock-in cells depleted of the piRNA factor Maelstrom (Mael), but not in unperturbed cells. Since Mael depletion causes strong transposon up-regulation, this has the potential to provide more substrate for the PICTS complex, therefore allowing capture of Nxf2-RNA interaction.

Murano et al. report binding of Nxf2 to transposon transcripts when using a stable line expressing myc-tagged Nxf2 coupled to knockdown of the endogenous protein. The authors compare all their Nxf2-CLIP data to a “size-matched input”. We have not attempted Nxf2-CLIP with this type of comparison.

Negative outcomes are obviously difficult to interpret, but we feel that our failure to detect enrichment of transposon transcripts in Nxf2-RIP could be due to differences in experimental protocols. We would like to highlight that we were able to reproduce the interaction of transposon transcripts with Panx by RIP-seq experiments (see Figure 4—figure supplement 1F), as reported (Sienski et al., 2015).

3) Discuss other known effectors such as Maelstrom, GTSF1 etc. and put this in the context of what we already know.

We have added a paragraph to the Discussion of our manuscript highlighting the other known TGS factors.

Murano et al. find that Panx interacts with Nxf2, Piwi, Mael and Arx (Murano et al., 2019, Supplementary Figure 1C). They furthermore show that myc-Nxf2 binds to Panx, Piwi, Nxt1 and Mael, but not Arx (Murano et al., 2019, Supplementary Figure 6C), which could mean that Mael and Arx do come into contact with the PICTS complex. However, the authors themselves do not discuss these findings (i.e. interaction with Mael) nor investigate this further. The other studies do not report such interaction (neither by coIP nor by IP-MS experiments), therefore further work will be necessary to understand the relationship between Mael, Arx and PICTS. We do not detect Piwi in our mass spectrometry experiment, so it is unlikely that we could detect the even less abundant Piwi interactor, Arx. We believe that our results along with the studies from the Brennecke, Siomi and Yu labs build a foundation for further examination and characterization of the hierarchy and composition of the piRNA-directed co-transcriptional silencing machinery.

4) Discuss the relationship between your results and those in *S. pombe*, showing that the nuclear mRNA export factor Mlo3 (human ALY/REF) opposes siRNA-mediated heterochromatin formation at mRNA genes (Yu et al., 2018). Your findings suggest that flies may have evolved a different mechanism that prevents piRNA targets/transposons, which act as templates for recruitment of downstream factors, from being exported.

We thank the reviewers for pointing out this study. We are now referencing it in our Discussion, highlighting these possibilities.